# Protocol for the development of a core outcome set for studies of pregnant women with pre-existing multimorbidity

Siang Ing Lee [1] , Kelly-Ann Eastwood [2,3] , Ngawai Moss [4]
Amaya Azcoaga-Lorenzo [5] , Anuradhaa Subramanian [1] , Astha Anand [1]
Beck Taylor [1] , Catherine Nelson-Piercy [6] , Christopher Yau [7]
Colin McCowan [5] , Dermot O'Reilly [2] , Holly Hope [8]
Jonathan Ian Kennedy [9] , Kathryn Mary Abel [8,10] , Louise Locock [11]
Peter Brocklehurst [1] , Rachel Plachcinski [4] , Sinead Brophy [9]
Utkarsh Agrawal [5] , Shakila Thangaratinam [12,13]
Krishnarajah Nirantharakumar [1] , Mairead Black [14]

K-AE, NM, ST, KN and MB contributed equally.

For numbered affiliations see end of article.

**Correspondence to**
Professor Krishnarajah Nirantharakumar;
k.nirantharan@bham.ac.uk

## ABSTRACT

**Introduction** Increasingly more pregnant women are living with pre-existing multimorbidity (≥two long-term physical or mental health conditions). This may adversely affect maternal and offspring outcomes. This study aims to develop a core outcome set (COS) for maternal and offspring outcomes in pregnant women with pre-existing multimorbidity. It is intended for use in observational and interventional studies in all pregnancy settings.

**Methods and analysis** We propose a four stage study design: (1) systematic literature search, (2) focus groups, (3) Delphi surveys and (4) consensus group meeting. The study will be conducted from June 2021 to August 2022. First, an initial list of outcomes will be identified through a systematic literature search of reported outcomes in studies of pregnant women with multimorbidity. We will search the Cochrane library, Medline, EMBASE and CINAHL. This will be supplemented with relevant outcomes from published COS for pregnancies and childbirth in general, and multimorbidity. Second, focus groups will be conducted among (1) women with lived experience of managing pre-existing multimorbidity in pregnancy (and/or their partners) and (2) their healthcare/social care professionals to identify outcomes important to them. Third, these initial lists of outcomes will be prioritised through a three-round online Delphi survey using predefined score criteria for consensus. Participants will be invited to suggest additional outcomes that were not included in the initial list. Finally, a consensus meeting using the nominal group technique will be held to agree on the final COS. The stakeholders will include (1) women (and/or their partners) with lived experience of managing multimorbidity in pregnancy, (2) healthcare/social care professionals involved in their care and (3) researchers in this field.

**Ethics and dissemination** This study has been approved by the University of Birmingham's ethical review committee. The final COS will be disseminated through peer-reviewed publication and conferences and to all stakeholders.

### Strengths and limitations of this study

► Core outcome set (COS) development in accordance to the COS standards for development.
► Extensive patient, public and stakeholder involvement at each stage.
► Pragmatic design to make the COS development feasible in the context of multimorbidity.
► The applicability of the COS may be limited to high-income countries.
► Responder bias may influence the types of outcomes included in the final COS.

## BACKGROUND

Multimorbidity is a state of having two or more long-term physical or mental health conditions.[1] Despite an increase in multimorbidity within the general population,[2] there is sparse literature for pregnant women with multimorbidity. Studies in the USA have reported that between 0.8% and 13.9% of hospital births were from women with multiple chronic conditions.[3 4] Using a list of 79 chronic conditions, our preliminary study found that one in four pregnant women in the UK had active multimorbidity at conception.[5]

Studies have shown that multimorbidity is associated with increased risk of adverse obstetric outcomes (eg, preterm birth) and severe maternal morbidities as a consequence of childbirth (eg, hysterectomy and eclampsia).[3 4] The 2020 UK national maternal mortality review reported that 90% of women who died within a year of pregnancy had multiple health and social problems.[6] The

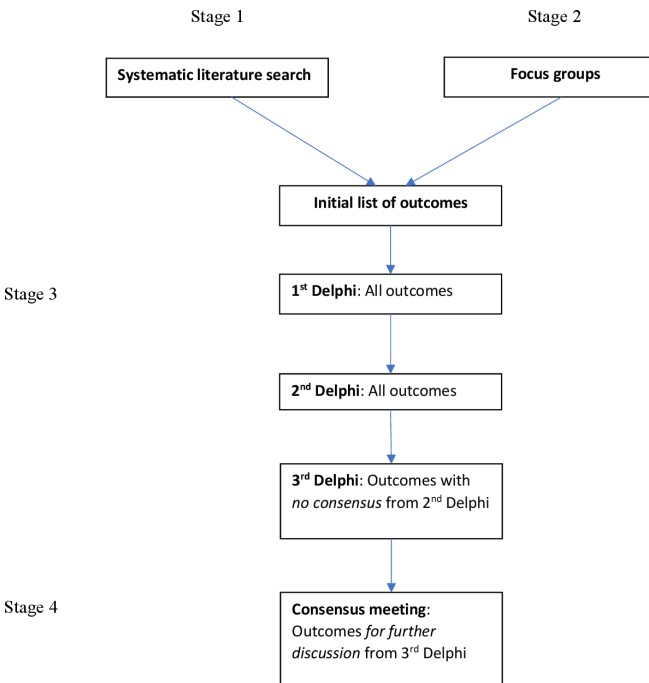

**Figure 1** Flowchart of a core outcome set development method.

leading direct cause of maternal death included thrombosis, thromboembolism and maternal suicide; leading indirect cause of death included cardiac diseases, epilepsy and stroke.[6] In addition to acute complications (eg, eclampsia) and chronic complications (progression from gestational diabetes to type II diabetes) for the mother, evidence suggests that pre-existing maternal morbidities and medications taken for these morbidities can lead to offspring complications such as neurodevelopmental disorders and congenital anomalies.[4 7–10] Current observational evidence and interventions focus on single morbidities. There is an urgent need for further understanding of the consequence of pre-existing maternal multimorbidity and development of interventions to improve maternity care for these women.[11 12]

To facilitate future research studies, a core outcome set (COS) is required. This will standardise the outcomes being reported, allow for evidence synthesis and ensure outcomes important to women, their families, carers and health and social care professionals are captured.[13] The importance of COS in women's health is endorsed by the Core Outcomes in Women's Health initiative.[14] The Core Outcome Measures in Effectiveness Trial (COMET) initiative collates resources for COS development and maintains a COS database.[15]

A recent scoping review identified 26 COSs relevant to maternity service users, of which 3 were related to pre-existing maternal morbidities in pregnancy (diabetes, epilepsy and infertility).[16] A search for COS in pregnancy on the COMET database further identified two published COS (depression and rheumatological conditions) and three in progress (cardiac disease, venous thromboembolism and immune thrombocytopenia).[15] There is

currently no COS for multimorbidity in pregnancy. We propose a pragmatic study design to develop a COS for observational and interventional studies, for pregnant women with pre-existing multimorbidity, covering obstetrics and maternal and offspring outcomes.

## METHODS

This study is designed in accordance with the COS standards for development Core Outcome Set-STAndardised Protocol (COS-STAD) recommendations and the protocol follows the COS-STAP statement (online supplemental appendix 1); study findings will be reported following the COS standards for reporting.[17–19] The planned start and end dates for the study are June 2021 and August 2022, respectively. The study is registered on the COMET database.[20]

The study will consist of four stages: (1) systematic literature search for reported outcomes for mother and child in studies of pregnant women with multimorbidity; (2) focus groups of women with lived experience of managing pre-existing multimorbidity in pregnancy and/or their partners, and their healthcare/social care professionals; (3) Delphi surveys among stakeholders to prioritise the core outcomes and (4) a consensus meeting to agree on the final COS (figure 1).

### Scope of the COS

The population is pregnant women; the exposure is pre-existing multimorbidity, defined as having two or more long-term physical or mental health conditions at conception.[1] This does not include pregnancy-related morbidities (eg, gestational diabetes), which will be considered as pregnancy outcomes. The morbidities do not have to be independent of each other. For instance, if a morbidity is a consequence of another morbidity (eg, diabetic eye disease and diabetes), these will be classed as two separate morbidities. The COS will be applicable principally to observational studies but will also inform interventional studies for pregnancy in all settings.

Maternal outcomes will include the antenatal, intrapartum and postpartum period. Offspring outcomes will include the neonatal (first 1 month), infant (first 1 year), prepubertal (2–11 years old), pubertal period (12–18 years old) and adulthood.[21] We have included outcomes across the lifespan of the offspring to inform observational studies that take a life-course approach.[22] Evidence is emerging that pre-existing maternal morbidities can impact on offspring long-term health in early adulthood.[23] Pregnancy outcomes in the rest of this protocol will refer to both maternal and offspring outcomes.

### Patient and public involvement

This protocol has been shaped by extensive patient and public involvement (PPI). PPI for this study will be three-tiered: (1) patient representatives in the scientific advisory group (SAG), (2) PPI advisory group and (3) patient and public stakeholders as research participants.

**Figure 2** Description of patient and public involvement in the core outcome set development.

The SAG consists of clinicians (specialists in maternal and fetal medicine, obstetrics, perinatal mental health, general practice and public health), researchers and women representatives collaborating on a larger project studying pregnant women with multimorbidity (MuM-PreDiCT).[24] NM, a women representative from the SAG, has advised on the study design, co-authored this protocol and created figure 2 that illustrates the PPI in the COS development.[25]

### Stage 1: systematic literature search

A pragmatic approach to identifying a list of initial outcomes will be adopted given the wide range of potential multimorbidities. We will first identify outcomes from published COS for pregnancy and childbirth and published COS for multimorbidity from the COMET database.[26–29] We will then conduct a systematic literature search for reported outcomes in published studies of pregnant women with multimorbidity.

### Search strategy

The following databases will be searched: Cochrane library, Medline, EMBASE and CINAHL. Relevant key search terms will include pregnancy (population and maternal outcomes), multimorbidity (exposure) and offspring (offspring outcomes) derived from previous literature.[28 30 31]

### Study selection and data extraction

The inclusion criteria are: systematic reviews, interventional studies, observational studies, qualitative studies and patient-reported outcome measures (PROM) studies; studies reporting pregnancy, maternal and offspring outcomes; and studies of pregnant women with multimorbidity. The exclusion criteria are ongoing studies with no published outcomes, editorials, commentaries, narrative reviews, case reports, case series, diagnostic accuracy studies, laboratory studies and animal studies. No time or language limits will be applied. Full text screening will be conducted by two independent reviewers.

Two reviewers will extract the following data from included studies: author, year of publication, study design, PROM domains, types of outcomes, definition of and measurement tools for the outcomes. Any discrepancy between the two independent reviewers for study selection and data extraction will be resolved with a third reviewer.

### Stage 2: focus groups

Outcomes identified in the published literature may represent outcomes considered as important to researchers.[13] Therefore, focus groups will be conducted to ensure the capture of outcomes considered as important to women with lived experience of managing pre-existing multimorbidity in pregnancy and/or their carers/partners (two focus groups), and healthcare/social care professionals involved in their care (one focus group). The synergistic discussion in focus groups will allow participants to consider outcomes which are important to others and stimulate in-depth discussions.[32]

We will aim to include six–eight participants per focus group. Sampling will be purposive and guided by the sampling matrix to provide a broad representation of

**Table 1** Sampling matrix for the focus groups, Delphi surveys and consensus meeting

| Characteristics | Target/minimum numbers* | | |
| --- | --- | --- | --- |
| | Focus groups | Delphi surveys[47] | Consensus meeting |
| (1) Women with lived experience of managing pre-existing multimorbidity (two or more long-term conditions) in pregnancy | 12–16 | 50 | 5 |
| Physical health conditions | 6 | 8–10 | 1 |
| Mental health conditions | 3–6 | 8–10 | 1 |
| Ethnic minority | 3–6 | 8–10 | 2 |
| Socioeconomically disadvantaged/marginalised groups (eg, homeless, refugee, asylum seeker, drug and alcohol service users, disabled people or victims of domestic abuse)[6] | 3–6 | 8–10 | 1 |
| (2) Healthcare/social care professionals | 6–8 | 50 | 5 |
| Obstetric medicine/maternal medicine | 1–2 | 8–10 | 1 |
| Obstetric | 1–2 | 8–10 | 1 |
| Midwifery/antenatal practitioner | 1–2 | 8–10 | 1 |
| Perinatal mental health | 1–2 | 8–10 | 1 |
| Other: for example, primary care, public health, neonatologist, paediatrician, health visitor, commissioner, maternity service provider, social worker, drug and alcohol service provider and maternity advocate/educator | 2 | 8–10 | 1 |
| (3) Researchers Academics, triallist and journal editors (as future implementers) | – | 5–10 | 2 |

*NB: Target/minimum numbers are estimates. Due to the overlap of characteristics between participants (eg, physical and mental health conditions, healthcare/social care professionals and researchers), we will continuously review the characteristics of participants so that we can identify any underrepresented groups and target recruitment efforts in these areas.

stakeholders and characteristics (table 1). Recruitment channels are listed in table 2. Involvement of the underserved population will be guided by our PPI advisory group and the MuM-PreDiCT group's strategy for diverse representation.[5]

Based on the advice of our PPI advisory group, the focus groups will be held virtually. Participants will be sent participant information sheets in advance of the meeting and consent will be taken 24 hours later either in electronic form or verbally. The focus group will last for 90 min or until no further new ideas are forthcoming. A topic guide will be developed based on previous literature, and with the guidance of qualitative experts and patient representatives in the SAG and our PPI advisory group.[33 34] The focus group will be facilitated by a researcher with qualitative methodology training. The focus group discussion will be recorded using the virtual meeting platform, the recordings will be transcribed and imported to NVivo. Data analysis will be inductive, following a structured, multistage approach to thematic analysis.[35]

### Initial list of outcomes
The initial list of outcomes generated from stages 1 and 2 will be reviewed and refined by the SAG and PPI advisory group to combine outcomes that are clinically and pathophysiologically similar to avoid redundancy.[13 36] Pregnancy outcomes will be categorised by: (1) maternal or offspring outcomes and (2) by an established taxonomy of outcomes (mortality/survival, physiological/clinical,

life impact/functioning, resource use and adverse events/effects).[37]

### Stage 3: Delphi surveys
The Delphi technique collates stakeholder opinions using sequential surveys. The response is summarised and fed back to stakeholders anonymously in subsequent rounds. Stakeholders consider the collective views before re-rating the outcomes. This provides a mechanism to reconcile different opinions to reach a consensus.[13] This study will employ a three-round Delphi survey, which is generally sufficient to reach consensus (figure 1).[38] Participants will have the opportunity to suggest additional outcomes that were not included in the initial list.

The surveys will be hosted on a secure platform online. The three groups of stakeholders that will be invited to participate and the recruitment channels are outlined in table 2. There is no recommended sample size for Delphi surveys; instead of basing the sample size on statistical power, this is often a pragmatic choice.[13] Previous obstetric COS has achieved sample size of around 20–40 for patients and 50–100 for healthcare professionals.[36 39–41] To reach the target sample size, snowballing recruitment will be encouraged. To check for representation, the survey will ask for participant characteristics, including types of long-term conditions constituting multimorbidity, age, ethnicity, education level and socioeconomic status (patient representatives, as outlined in table 1), specialty and job roles (healthcare professionals

**Table 2** Stakeholders and recruitment channels

| Stakeholder group | Potential recruitment channels[48 49] |
|---|---|
| (1) Patient representatives<br>Women with lived experience of managing pre-existing multimorbidity (two or more long-term physical or mental health conditions) in pregnancy and/or their partners/carers | ▶ Service user associations/groups: for example, Maternity Voice Partnership<br>▶ Parent support networks: for example, National Childbirth Trust<br>▶ Community groups: local maternity groups, baby/toddler groups, local authority baby class, nursery, health visitor society, faith group and baby groups by church<br>▶ Social media: Facebook, Twitter, Instagram and Linkedin<br>▶ Parent-oriented social media: home schooling, weaning, budget family menu sites, breast feeding, outdoor activities for family, local outdoor groups, Mumsnet and Gingerbread (single parents)<br>▶ Patient support groups/charities for specific conditions: Tommy's, Epilepsy Action, Association of Medical Research UK member charities and National Council for Voluntary Organisations<br>▶ Royal Colleges women's networks: Royal College of Obstetrics and Gynaecology Women's Voices Involvement Panel, and Royal College of Midwifery Maternity Voices Network<br>▶ Victims of domestic abuse: Refuge, Women's Aid, WE:ARE (Women's Empowerment and Recovery Educators)<br>▶ People with disability: Disabled Parents Network, disabled parents Facebook groups<br>▶ Drug and alcohol: Drug and Alcohol Abuse Support for Women<br>▶ Refugee: Refugee Council, Refugee Survival Trust<br>▶ LGBT: LGBT Mummies Tribe, Stonewall, Facebook groups for transgender men or lesbian women experiencing pregnancy |
| (2) Healthcare/social care professionals<br>Any healthcare/social care professionals involved in providing multidisciplinary team care for pregnant women: for example, obstetric physicians, obstetricians, physicians, paediatricians, neonatologists, psychiatrists, primary care clinicians, public health professionals, clinicians of established joint antenatal clinics, perinatal mental health team, drug and alcohol services, social services, midwives, health visitors, dieticians, policy-makers and commissioners | ▶ Personal, professional and clinical network of the researchers<br>▶ Royal Colleges<br>▶ Societies (eg, McDonald Obstetric Medicine Society, European Board and College of Obstetrics and Gynaecology)<br>▶ Maternity charities (eg, Ammalife and Elly)<br>▶ Social media for professional groups (eg, Twitter and Facebook). |
| (3) Researchers<br>Academics, triallist and journal editors (as future implementers) | The SAG's personal network, social media (Twitter), the COMET and Core Outcomes in Women's Health (CROWN) network, the Cochrane Pregnancy and Childbirth group, and peer-reviewed journals of obstetric medicine and obstetrics |

COMET, Core Outcome Measures in Effectiveness Trial; SAG, scientific advisory group.

and researchers). Participant's name and email contact will be included to avoid duplicate entry, for sending up to 2 personalised reminders (1 week apart) and following up on incomplete response. This information will be kept securely, confidentially and separate from the survey responses.

Care will be taken in explaining the concept of COS to lay participants, using supporting materials from the COMET website.[15] The wording of the survey will be developed using appropriate language commonly used by representatives in the focus groups. The SAG and PPI advisory group will also ensure that plain language is used to describe the outcomes of interest. Outcomes will be presented in alphabetical order to avoid any response effects related to the order of survey items.[13 42]

Each outcome will be rated on a 9-point Likert scale: 1–3 (not important), 4–6 (important but not critical) and 7–9 (critically important). An 'unable to score' option will be provided to allow for participants who may not have the expertise to score certain outcomes.[13] The 9-point Likert scale is commonly used in COS studies and recommended by the Grading of Recommendations Assessment, Development and Evaluation Working Group.[13 43]

### Score criteria for consensus
▶ Consensus in is when ≥70% of all participants rated 7–9 (critically important) for an outcome.
▶ Consensus out is when≥70% of all participants rated 1–3 (not important) for an outcome.
▶ No consensus is for any other scores.

► For further discussion is when: (1) ≥70% of all participants rated 4–6 (important but not critical) for an outcome, or (2) when ≥70% of patient representatives have rated 7–9 for an outcome but consensus in is not reached.[44]

## Pilot study

The survey will be piloted before the Delphi rounds to check face validity. It will also inform the time frame required for completion of each Delphi round.

## First Delphi

Participants will be sent a participant information sheet explaining the objectives of the COS study. Completion of the online survey assumes implied consent. Participants will be informed that they can withdraw their response from the study within 1 week of submitting the survey. Once the name and contact details are separated from the survey response, it will not be possible to withdraw their survey response.

At the end of the survey, an open question will invite participants to suggest a maximum of two additional outcomes. If a new outcome is suggested by two or more participants, it will then be added to the second Delphi round. Depending on how many new outcomes that will be presented, this criterion may be modified on a pragmatic basis.

## Secound Delphi

Participants who responded to the first Delphi round will be invited to participate in the second Delphi. A summary response from the first Delphi stratified by stakeholder groups will be presented for all outcomes.

## Third Delphi

Participants who responded to the second Delphi round will be invited to participate in the third Delphi. Outcomes that reached no consensus will be included as options in the third Delphi survey. A summary response from the second Delphi round stratified by stakeholder groups will also be presented. Attrition rate will be calculated for each subsequent rounds.

## Stage 4: consensus meeting

At the time of writing, the UK is undergoing social distancing due to the COVID-19 pandemic. In addition, our SAG patient representative has advised that travelling to meetings may not be convenient for mothers with childcare needs. Therefore, the consensus meeting will be conducted through a virtual platform online.

The consensus meeting panel will be purposefully selected from the SAG, PPI advisory group and Delphi survey respondents to ensure representation of a range of backgrounds. In the third Delphi survey, participants will be asked about their willingness to attend the consensus meeting. For meaningful engagement in the consensus meeting, we will aim for 10–15 participants.[13 25 42]

An experienced facilitator will be the non-voting chair. Summary scores stratified by stakeholder groups will be

presented for outcomes that met the 'for further discussion' criteria. Nominal group technique will be used to discuss these outcomes.[44 45] Participants will be asked to contemplate independently whether these outcomes should be included. Each participant will be invited to voice their reasoning in turn using a round-robin format to avoid domination of the discussion by selected few. This will be followed by an open discussion, after which a final anonymous binary vote of yes/no will be conducted for each of these outcomes. Outcomes that received ≥70% yes votes will be included in the final COS.

## DISCUSSION

The proposed COS will be applicable for observational and interventional studies for pregnant women with pre-existing multimorbidity. Further interventional studies are urgently needed to tackle multimorbidity in pregnancy and reduce the associated adverse outcomes. It is, therefore, important to have a predefined COS to inform future research studies to enable valid comparisons between study findings.

## Strength

There is currently no COS for studies of pregnant women with multimorbidity. As multimorbidity covers a wide range of diseases, this presents a unique methodological challenge to the COS development. This study aims to adopt a pragmatic approach to make the task manageable while still following the COS-STAD minimum standards. Inclusion of observational studies in generating the initial list of outcomes may detect rare but important clinical outcomes especially for offspring.[46]

The Delphi surveys, nominal group technique and anonymous final vote in the consensus meeting will encourage participation of all stakeholders and avoid dominance of selected figures. As outlined in figure 2, PPI will have a meaningful role throughout the COS development to ensure accessibility and relevance to patient stakeholder groups and that patient perspectives are represented in the governance of the COS development.[25]

To widen its applicability, the proposed COS will include both maternal and offspring outcomes and will include outcomes that are common to all pregnant women with multimorbidity. Finally, by creating this COS, we hope to encourage and facilitate urgently needed research for pregnant women with multimorbidity.

## Limitation

The focus groups, Delphi survey and consensus meeting will be conducted in English. Although efforts will be made to encourage international participation, this may limit the generalisability of the findings to high-income countries. The use of online platforms may lead to under-representation of the digitally disadvantaged groups. Similarly, responder bias may influence the types of outcomes included in the final COS. To ensure representation of the socially disadvantaged/marginalised

group and healthcare/social care professionals with busy work schedules, our approach will be flexible and where necessary/preferred by the participants, we will offer the option of one-to-one interviews instead of focus groups.

As further epidemiological knowledge is gained in identifying common morbidity clusters in pregnant women, the COS may need to be updated to incorporate outcomes specific to these clusters.

## DISSEMINATION

The final COS will be fed back to all stakeholders. Patient and public representatives will be encouraged and supported to share the difference they have made. With the guidance of the SAG and the PPI advisory group, a collaborative dissemination plan will be formulated. This will include submitting the findings for publication in a peer-reviewed journal, dissemination at conferences and registering the study on the COMET database.

**Author affiliations**
[1]Institute of Applied Health Research, University of Birmingham, Birmingham, UK
[2]Centre for Public Health, Queen's University Belfast, Belfast, UK
[3]St Michael's Hospital, University Hospitals Bristol and Weston NHS Foundation Trust, Bristol, UK
[4]Patient and Public Representative, London, UK
[5]School of Medicine, University of St Andrews, St Andrews, UK
[6]Guy's and St Thomas' NHS Foundation Trust, London, UK
[7]Division of Informatics, Imaging and Data Sciences, Faculty of Biology, Medicine and Health, The University of Manchester, Manchester, UK
[8]Division of Psychology and Mental Health, Faculty of Biology Medicine and Health, Centre for Women's Mental Health, The University of Manchester, Manchester, UK
[9]Data Science, Medical School, University of Swansea, Swansea, UK
[10]Greater Manchester Mental Health NHS Foundation Trust, Manchester, UK
[11]Health Service Research Unit, University of Aberdeen, Aberdeen, UK
[12]WHO Collaborating Centre for Global Women's Health, Institute of Metabolism and Systems Research, University of Birmingham, Birmingham, UK
[13]Department of Obstetrics and Gynaecology, Birmingham Women's and Children's NHS Foundation Trust, Birmingham, UK
[14]School of Medicine, Medical Science and Nutrition, University of Aberdeen, Aberdeen, UK

**Contributors** Our authors list includes PPI co-investigators NM and RP. SIL, K-AE, NM, AA-L, AS, AA, BT, CN-P, CY, CM, DOR, HH, JK, KMA, LL, PB, RP, SB, UA, ST, KN and MB conceived the study, contributed to the study design, critically reviewed and revised the protocol drafts, agreed on the final draft manuscript for submission and are accountable for all aspects of the work. SIL led the development of the protocol and drafted the initial manuscript with contribution and supervision from KN, ST, MB, K-AE. LL, BT and MB contributed to the qualitative element of the study design and, together with NM, RP, SB and KMA, advised on the recruitment channels; PPI co-investigator NM designed figure 2.

**Funding** This work was supported by Medical Research Council consolidator grant (grant number: MR/V005243/1).

**Competing interests** None declared.

**Patient consent for publication** Not required.

**Provenance and peer review** Not commissioned; externally peer reviewed.

**ORCID iDs**
Siang Ing Lee http://orcid.org/0000-0002-2332-5452
Kelly-Ann Eastwood http://orcid.org/0000-0003-3689-0490
Ngawai Moss http://orcid.org/0000-0001-9369-5072
Amaya Azcoaga-Lorenzo http://orcid.org/0000-0003-3307-878X
Anuradhaa Subramanian http://orcid.org/0000-0001-8875-7363
Astha Anand http://orcid.org/0000-0003-0494-321X
Beck Taylor http://orcid.org/0000-0002-3559-7922
Catherine Nelson-Piercy http://orcid.org/0000-0001-9311-1196
Christopher Yau http://orcid.org/0000-0001-7615-8523
Colin McCowan http://orcid.org/0000-0002-9466-833X
Dermot O'Reilly http://orcid.org/0000-0002-9181-0652
Holly Hope http://orcid.org/0000-0002-4834-6719
Jonathan Ian Kennedy http://orcid.org/0000-0002-1122-6502
Kathryn Mary Abel http://orcid.org/0000-0003-3538-8896
Louise Locock http://orcid.org/0000-0002-8109-1930
Peter Brocklehurst http://orcid.org/0000-0002-9950-6751
Rachel Plachcinski http://orcid.org/0000-0001-9908-0773
Sinead Brophy http://orcid.org/0000-0001-7417-2858
Utkarsh Agrawal http://orcid.org/0000-0001-5181-6120
Shakila Thangaratinam http://orcid.org/0000-0002-4254-460X
Krishnarajah Nirantharakumar http://orcid.org/0000-0002-6816-1279
Mairead Black http://orcid.org/0000-0002-6841-8601

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
