## [Reviewer comments · BMJ Open]

ARTICLE DETAILS

TITLE (PROVISIONAL)	Protocol for the development of a core outcome set for studies of pregnant women with pre-existing multimorbidity
AUTHORS	Lee, Siang Ing; Eastwood, Kelly-Ann; Moss, Ngawai; Azcoaga-Lorenzo, Amaya; Subramanian, Anuradha; Anand, Astha; Taylor, Beck; Nelson-Piercy, Catherine; Yau, Christopher; McCowan, Colin; O'Reilly, Dermot; Hope, Holly; Kennedy, Jonathan; Abel, Kathryn; Locock, Louise; Brocklehurst, Peter; Plachcinski, Rachel; Brophy, Sinead; Agrawal, Utkarsh; Thangaratinam, Shakila; Nirantharakumar, Krishnarajah; Black, Mairead

VERSION 1 – REVIEW

REVIEWER	Hewawasam, Erandi Women's and Children's Hospital Adelaide, Department of Endocrinology and Diabetes
REVIEW RETURNED	01-Dec-2020

GENERAL COMMENTS	This is an important study to develop core outcome set for studies of pregnancy affected by pre-existing multimorbidity. This is a well written study protocol with clear aims and study methods. Please address/clarify the following points to further strengthen your study protocol. 1. The short title- Not the most appropriate. Please consider rewording it to show that this is a protocol for developing COS.2. Abstract- Please move the aim (lines 17-19) to the introduction section3. Abstract- Start the Methods and analysis with the study design/set-up "This is a"4. Strengths and Limitations-You may list the potential biases that may arise in this study as a limitation5. Background section is a bit on the shorter side for a protocol study. It would be great if the authors could expand on some of the points listed in the background eg. the rationale for the study, more examples of short term and long term outcomes of pregnancies with multimorbidity6. Please pay attention to spelling eg: line 42 "whist"
---

REVIEWER	D'Souza, Rohan University of Toronto Lead Investigator of the Outcome Reporting in Obstetric Studies Initiative.
-----------------	--

REVIEW RETURNED	02-Feb-2021
-----------------	-------------

GENERAL COMMENTS	Dear authors, Although this protocol provides a general overview for the development of a core outcome set, it lacks some important specifics. In addition, there are a number of other issues that require clarification, as listed below. Abstract  - Pre-existing multimorbidity needs to be defined - A brief description of the nature of the scientific advisory group that identified these multi-morbidities would need to be provided - The abstract suggests that the hierarchical systematic review will be followed by a Delphi process, while forgoing the step of qualitative interviews with patients. While, the systematic review may include patient-reported outcome studies, what provisions have been put in place, if there are no such studies conducted with the aim of eliciting outcomes? - Given that the Delphi process will be conducted online, the rationale for limiting stakeholders to the UK is unclear, as their values and preferences may not represent those from other high-resource settings, and will completely exclude those from middle- and low-resource settings that are also affected by multimorbidities and where a large proportion of trials in the area would be conducted. Introduction  - The term 'multimorbidity' needs to be described more clearly, either here or in Stage 1 of the methods. The SAG prioritized 90 morbidities. By multimorbidity, do the authors mean a study involving participants with two or more morbidities from this list of 90? Do these morbidities need to be independent of each other; if not, what degree of overlap between conditions would qualify as multimorbidity? - Suggest that the scope of the COS is clarified in the introduction, without which the statement that this COS is intended to be used for 'all clinical research' presents more questions than answers. Methods  - Suggest describing the stages in order, and including the description of stakeholders under Stage 2. - A major flaw is the lack of a step specifically designed to elicit the patient perspective. Many recent COS in in obstetrics have identified the need for inclusion of this step, as qualitative research conducted in areas, often focus on experiences and not outcomes. Merely synthesizing the qualitative literature is unlikely to elicit what outcomes pregnant persons and stakeholders consider important. - With regard to the Delphi method, while the authors indicate that they will ensure diversity of the sample, no sampling strategy has been presented. A clear sampling strategy which identifies what the authors mean by a diverse sample would be required for e.g. representation of adequate numbers of participants with different types of physical and mental morbidities, disenfranchised groups, varied socio-economic strata, cultural groups, etc.
------------------	--

- The list of 90 morbidities identified by the SAG would need to be presented somewhere, for readers/reviewers to better understand the scope of what is being studied.
- The first paragraph of Stage 1 is vague and would need to be clarified further to outline what the hierarchical systematic search will entail.
- By published COS, do the authors mean a COS of a condition in pregnancy or the condition in general. For e.g. 'SLE in pregnancy' or 'SLE'? I presume the former, given the reference cited.
- Similarly, not all systematic reviews are conducted with the intent of identifying outcomes, By 'systematic reviews of observational/interventional studies', do the authors mean, 'systematic reviews conducted with the intention of developing COS, if the COS is not yet published'? The references seem to suggest otherwise. Including outcomes reported in systematic reviews such as those referenced will not generate a comprehensive list of outcomes.
- How will the authors address a plausible situation where there are no patient-reported outcome studies? Where will patient reported outcomes be elicited from?
- Search strategy: While the COMET database is likely to identify many of the relevant studies, the Cochrane Library does not often include systematic reviews aimed at eliciting outcomes, and Medline is insufficient to identify patient-reported outcome studies, many of which are published in journals not indexed on Medline.
- Although sample searches have been provided, it is not immediately clear whether the process is going to be undertaken for each of the 90 identified morbidities, and if not, how they will be clustered into relevant groups.
- Would the authors be able to present, based on their preliminary searches, how many of the morbidities they think COS already exist for, which will testify to the feasibility of the project.
- It is not clear whether Delphi participants will be presented with a list of outcomes common to all conditions in addition to 90 (or fewer) specific lists of outcomes. The burden of having to respond to outcomes related to a single morbidity is immense. How do the authors hope to address the long list of outcomes from so many conditions?
- The authors would need to describe the process of condensing outcomes from the long list, removal of redundancies, and dealing with outcomes common to two or more specific conditions but not to all. Would these outcomes be repeated for each cluster?
- What do the authors mean by '(consensus criteria) will be applied to the aggregate scores for all participants stratified by stakeholder group'?
- In addition to the above, the following issues need to be addressed
 - o Sample size and justification
 - o Sampling criteria for pregnant persons, clinicians and researchers – as well as relative proportions
 - o Details on comorbidities/ clusters of comorbidities and the relative involvement of specialists e.g. would a nephrologist be exempt from answering questions related to cardiology or general obstetrics, or a neonatologist be exempt from responding to questions on any of the maternal medical morbidities?
- Consensus meeting

	o Sample size, representation and justification -especially important given the diverse range of conditions, some of which are going to be irrelevant to a large proportion of participants. Will there be patient representation for each condition? One patient representative for all conditions? What about specialists? o What (qualitative) approach will be used to arrive at consensus? The description provided is vague. Discussion  - The clinical and research application would need to be defined more clearly. - Inclusion of stakeholders from one country and the serious implications on its generalizability thereof, would need to be addressed as a limitation - The lack of a qualitative step to specifically elicit outcomes especially given the limited search for patient reported outcomes and the lack of qualitative studies aimed at eliciting outcomes in this population, would need to be stated as a serious limitation Reference list  - The references are not reflective of the work done in the area of core outcome sets for pregnancy-associated conditions.
--	---

REVIEWER	Shah, Vibhuti Mount Sinai Hospital, Paediatrics
REVIEW RETURNED	03-Feb-2021

GENERAL COMMENTS	Thank you for the opportunity to review for publications. The authors/investigators plan to develop a Core Outcome Set for maternal and offspring outcomes in pregnant women with pre-existing multimorbidity. The authors/investigators state the plan in regards to development of the COS using a 3 step process. Issues:  1) Abstract: The dates of the study are not stated in the protocol (a requirement for BMJ Open). 2) Background section:  a) Line 2 the authors should provide the burden of illness, i.e. what is the incidence or rate of multimorbidity issues in the pregnant woman? b) Line 6: It is stated that 80% of maternal deaths occur in multimorbidity- what are the 3 or 5 most common causes of death and is there any intervention that can be offered to reduce mortality? c) The 4th paragraph in the background section is all about the "Significance of the study" and not sure if this should be part of this section. More relevant for the discussion section. 3) Methods:  a) Scope of the COS- It is stated that long-term outcomes will be included? What does long-term mean (in regards to time frame)? Will such data be available as the outcomes change over time with advances in science? b) For the offspring outcomes- the authors have made distinction between newborn (first 7 days of life) and neonate (first 28 days of life). However, in medical contexts, newborn or neonate refers to
---

	an infant in the first 28 days after birth. The authors need to justify this distinction. Also they are evaluating long-term outcomes of offspring. Can the authors justify or provide the rationale for assessing outcomes across the life span for both mother and her offspring? c) What would be the sample size for each of the stakeholder groups? d) What is the rationale for including journal editors in the stakeholder group? e) The number of stakeholders involved for each group not described? How will be participant bias/response bias be addressed? How will discrepancies be addressed? Recognizing that there will be potential differences between patient vs. physician preferences or importance in terms of outcomes, how will this issue be addressed? f) The authors talk about diversity of sample- will patients be selected from different ethnic groups? Will education and SES impact selection of outcomes? g) The time line for literature search has not been described? It appears that there is an iterative process and search will be considered until no unique outcomes is identified. Do the authors have an idea of how many years of literature search would have to be conducted based on their previous experience? h) The authors list some of the databases that will be searched. Why is Embase excluded? Why limit to English language literature? i) Initial list of outcomes provide description of pregnancy outcomes. How will offspring outcomes be reviewed?
--	---

VERSION 1 – AUTHOR RESPONSE

Reviewer 1		
1	This is an important study to develop core outcome set for studies of pregnancy affected by pre-existing multimorbidity. This is a well written study protocol with clear aims and study methods. Please address/clarify the following points to further strengthen your study protocol.	We thank Reviewer 1 for the encouraging comments.
2	The short title- Not the most appropriate. Please consider rewording it to show that this is a protocol for developing COS.	The title has been changed to: 'Protocol for the development of a core outcome set for studies of pregnant women with pre-existing multimorbidity'

3	Abstract- Please move the aim (lines 17-19) to the introduction section	The following sentence has been moved to the Introduction section in the abstract as advised. “This study aims to develop a COS for maternal and offspring outcomes in pregnant women with pre-existing multimorbidity.”
4	Abstract- Start the Methods and analysis with the study design/set-up “This is a	We have added the study design at the start of the Methods section of the Abstract. “We propose a four stage study design: 1) systematic literature search, 2) focus groups, 3) Delphi surveys, and 4) consensus group meeting.”

5	Strengths and Limitations- You may list the potential biases that may arise in this study as a limitation	We have added the potential biases to the Strengths and Limitations section.  ● The applicability of the COS may be limited to high income countries ● Responder bias may influence the types of outcomes included in the final COS
---	--	--

6	Background section is a bit on the shorter side for a protocol study. It would be great if the authors could expand on some of the points listed in the background eg. the rationale for the study, more examples of short term and long term outcomes of pregnancies with multimorbidity	We have expanded the background section as advised and provided more examples of outcomes. We have also highlighted the lack of literature for pregnancies affected by pre-existing multimorbidity and used example outcomes of single morbidities to supplement this. “...there is sparse literature for pregnant women with multimorbidity” “...as studies have shown that multimorbidity was associated with increased risk of adverse obstetric outcomes (e.g. preterm birth) and severe maternal morbidities as a consequence of childbirth (e.g. hysterectomy, eclampsia).” “Besides acute complications (e.g. eclampsia) and chronic complications (progression from gestational diabetes to type II diabetes) for the mother, evidence suggest that maternal morbidities and medications taken for these morbidities can lead to offspring complications such as neurodevelopmental disorder and congenital anomalies” “Current evidence and interventions focus on single morbidities.” “There is currently no COS for pregnancy with multimorbidity.”
7	Please pay attention to spelling eg: line 42 “whist”	This has been corrected in Discussion: Strength section line 4 to ‘whilst’.
	Reviewer 2	
1	Although this protocol provides a general overview for the development of a core outcome set, it lacks some important specifics. In addition, there are a number of other issues that require clarification, as listed below.	We thank Reviewer 2 for the comments and have responded to the issues outlined.

2	Abstract: Pre-existing multimorbidity needs to be defined	The definition of pre-existing multimorbidity has been added to the Introduction section in the Abstract: “Increasingly more pregnant women are living with pre-existing multimorbidity (≥ 2 long-term physical or mental health conditions).”
3	Abstract: A brief description of the nature of the scientific advisory group that identified these multimorbidities would need to be provided	We have changed the study design to identifying the initial outcomes in studies of multimorbidity in general and will be developing a COS common for all pregnancies with multimorbidity. We will not limit the types of maternal morbidities for participants to the list of conditions our SAG has identified for the purpose of an epidemiological study. However, we have provided the reference to our study website in the Introduction section for more details of the SAG and the list of conditions.
4	Abstract: The abstract suggests that the hierarchical systematic review will be followed by a Delphi process, while forgoing the step of qualitative interviews with patients. While, the systematic review may include patient-reported outcome studies, what provisions have been put in place, if there are no such studies conducted with the aim of eliciting outcomes?	We thank Reviewer 2 for raising the issues of provision for if PROM studies are not identified in the literature search. We have added focus group as a qualitative component to the Methods section. In addition, we have made the provision to include outcomes that are important to our stakeholders, including women / their partners, by inviting them to suggest additional outcomes that may not be identified through the hierarchical literature search. This was outlined in the original Methods section: 1st Delphi and is now also added to the Abstract: “In the Delphi survey, stakeholders will be invited to suggest additional outcomes that were not included in the initial list.” The following has also been added to the Methods: Stage 2 Delphi Survey section to emphasise the point: “Participants will have the opportunity to suggest additional outcomes that were not included in the initial list.”

5	Abstract: Given that the Delphi process will be conducted online, the rationale for limiting stakeholders to the UK is unclear, as their values and preferences may not represent those from other high-resource settings, and will completely exclude those from middle- and low-resource settings that are also affected by multimorbidities and where a large proportion of trials in the area would be conducted.	We thank Reviewer 2 for the suggestion and have removed this from the Abstract and Methods section.
6	Introduction: The term 'multimorbidity' needs to be described more clearly, either here or in Stage 1 of the methods. The SAG prioritized 90 morbidities. By multimorbidity, do the authors mean a study involving participants with two or more morbidities from this list of 90? Do these morbidities need to be independent of each other; if not, what degree of overlap between conditions would qualify as multimorbidity?	We will not limit the types of maternal morbidities for participants to the list of conditions our SAG has identified for the purpose of a separate epidemiological study. We have defined multimorbidity as 'having two or more long-term physical or mental health conditions' in both Introduction and Methods: Scope of the COS section. We have also added the following statement to Methods: "The morbidities do not have to be independent of each other, e.g. if a morbidity is a consequence of another morbidity (e.g. diabetic eye disease and diabetes), these will be classed as two separate morbidities."

7	Introduction: Suggest that the scope of the COS is clarified in the introduction, without which the statement that this COS is intended to be used for 'all clinical research' presents more questions than answers.	We have clarified this further in the Background section and highlighted the evidence gap that requires further studies where the proposed COS is needed. "There is an urgent need for further understanding of the consequence of pre-existing maternal multimorbidity for pregnant women and their offspring and development of interventions to improve maternity care for these women." "..we propose a pragmatic study design to develop a COS for observational and interventional studies for pregnant women with pre-existing multimorbidity, covering obstetric, maternal and offspring outcomes."
8	Methods: Suggest describing the stages in order, and including the description of stakeholders under Stage 2.	We have moved the description of stakeholders to Stage 2: focus groups (Table 1 and 2).
9	Methods: A major flaw is the lack of a step specifically designed to elicit the patient perspective. Many recent COS in in obstetrics have identified the need for inclusion of this step, as qualitative research conducted in areas, often focus on experiences and not outcomes. Merely synthesizing the qualitative literature is unlikely to elicit what outcomes pregnant persons and stakeholders consider important.	We have now added focus groups as a qualitative component in Stage 2. This is described in the Methods section and we have also discussed the limitation of this in the Discussion section. We have also made the provision of inviting all stakeholders to suggest additional outcomes that they feel are important but have not been identified in the literature search in the Delphi survey. This is outlined in Methods: 1st Delphi section, and is now also emphasised in Abstract and Methods: Stage 2 Delphi survey section.

10	Methods: With regard to the Delphi method, while the authors indicate that they will ensure diversity of the sample, no sampling strategy has been presented. A clear sampling strategy which identifies what the authors mean by a diverse sample would be required for e.g. representation of adequate numbers of participants with different types of physical and mental morbidities, disenfranchised groups, varied socio-economic strata, cultural groups, etc.	Guided by reviewer's 2 literature, we have added a sampling matrix in Table 1.
11	Methods: The list of 90 morbidities identified by the SAG would need to be presented somewhere, for readers/reviewers to better understand the scope of what is being studied.	We will not limit the types of maternal morbidities for participants to the list of conditions our SAG has identified for the purpose of an epidemiological study. However, we have provided the reference to our study website in the Introduction section for the list of conditions (https://mumpredict.org/portfolio/shared-findings/).
12	Methods: The first paragraph of Stage 1 is vague and would need to be clarified further to outline what the hierarchical systematic search will entail.	We have changed the literature search strategy, so the hierarchical systematic search is no longer applicable.
13	Methods: By published COS, do the authors mean a COS of a condition in pregnancy or the condition in general. For e.g. 'SLE in pregnancy' or 'SLE'? I presume the former, given the reference cited.	We have changed the literature search strategy. For published COS, we have made it clear whether this is in pregnancy or in general. "We will first identify outcomes from published COS for pregnancy and childbirth in general and multimorbidity in general from the COMET database."

14	Methods: Similarly, not all systematic reviews are conducted with the intent of identifying outcomes, By ‘systematic reviews of observational/interventional studies’, do the authors mean, ‘systematic reviews conducted with the intention of developing COS, if the COS is not yet published’? The references seem to suggest otherwise. Including outcomes reported in systematic reviews such as those referenced will not generate a comprehensive list of outcomes.	Our literature search strategy has been kept broad by using the concept of (1) pregnancy (population and maternal outcomes), and (2) multimorbidity (exposure), we believe this would also capture systematic reviews conducted with the intention of developing COS in addition to the conventional observational/ interventional studies.
15	Methods: How will the authors address a plausible situation where there are no patient-reported outcome studies? Where will patient reported outcomes be elicited from?	We have now added focus group to elicit outcomes considered important to patients. Patients who participate in the Delphi surveys will also be invited to suggest additional outcomes. This is covered in the Abstract and Methods Delphi survey sections.
16	Search strategy: While the COMET database is likely to identify many of the relevant studies, the Cochrane Library does not often include systematic reviews aimed at eliciting outcomes, and Medline is insufficient to identify patient-reported outcome studies, many of which are published in journals not indexed on Medline.	We thank Reviewer 2 for the suggestion and have now added EMBASE and CINAHL to our Methods: Search strategy section: “The following databases will be searched: Cochrane library, Medline, EMBASE and CINAHL.”

17	Methods: Although sample searches have been provided, it is not immediately clear whether the process is going to be undertaken for each of the 90 identified morbidities, and if not, how they will be clustered into relevant groups.	We have changed the literature search strategy to cover multimorbidity overall in pregnancy and not the individual constituent morbidities.
18	Methods: Would the authors be able to present, based on their preliminary searches, how many of the morbidities they think COS already exist for, which will testify to the feasibility of the project.	We thank the Reviewer 2 for this suggestion. This has been added to the Introduction section and as a result we have changed the literature search strategy to cover multimorbidity overall in pregnancy and not the individual constituent morbidities. “A recent scoping review identified 26 COSs relevant to maternity service users, of which three were related to pre-existing maternal morbidities in pregnancy (diabetes, epilepsy, infertility). A search for COS in pregnancy on the COMET database further identified two published COS (depression, rheumatological conditions) and three in progress (cardiac disease, venous thromboembolism and immune thrombocytopenia).”
19	Methods: It is not clear whether Delphi participants will be presented with a list of outcomes common to all conditions in addition to 90 (or fewer) specific lists of outcomes. The burden of having to respond to outcomes related to a single morbidity is immense. How do the authors hope to address the long list of outcomes from so many conditions?	We thank Reviewer 2 for the advice and have changed the list of outcomes to be generic and applicable across all pregnancies with multimorbidity. When further epidemiological evidence is available for common or key morbidity clusters, further relevant COS can be developed then.

20	Methods: The authors would need to describe the process of condensing outcomes from the long list, removal of redundancies, and dealing with outcomes common to two or more specific conditions but not to all. Would these outcomes be repeated for each cluster?	The study is now designed to identify outcomes that are common to all pregnancies with multimorbidity. We have added some more details on the criteria for combining outcomes. “The initial list of outcomes generated from stage 1 and 2 will be reviewed and refined by the SAG and PPI advisory group to combine outcomes that are clinically and pathophysiologically similar to avoid redundancy.”
21	Methods: What do the authors mean by ‘(consensus criteria) will be applied to the aggregate scores for all participants stratified by stakeholder group’	We have now changed the consensus criteria to be applied for all participants.  ● Consensus in is when ≥70% of all participants rated 7-9 (critically important) for an outcome. ● Consensus out is when ≥70% of all participants rated 1-3 (not important) for an outcome. ● No consensus is for any other scores. ● For further discussion is when: (1) ≥70% of all participants rated 4-6 (important but not critical) for an outcome, or (2) when ≥70% of patient representatives have rated 7-9 for an outcome but consensus in is not reached.

22	Methods: In addition to the above, the following issues need to be addressed  o Sample size and justification o Sampling criteria for pregnant persons, clinicians and researchers – as well as relative proportions o Details on comorbidities/ clusters of comorbidities and the relative involvement of specialists e.g. would a nephrologist be exempt from answering questions related to cardiology or general obstetrics, or a neonatologist be exempt from responding to questions on any of the maternal medical morbidities? 	We have now provided sample size and sampling matrix in Table 1 and justification of the minimum sample size in Methods: Stage 3: Delphi survey section. We thank Reviewer 2 for highlighting the logistic challenges to limit the survey item burden for specialists and have changed our study design to include outcomes that are common to all pregnancies with multimorbidity. “There is no recommended sample size for Delphi surveys; instead of basing the sample size on statistical power, this is often a pragmatic choice. Previous obstetric COS has achieved sample size of around 20-40 for patients and 50-100 for health care professionals.”
23	Consensus meeting: Sample size, representation and justification -especially important given the diverse range of conditions, some of which are going to be irrelevant to a large proportion of participants. Will there be patient representation for each condition? One patient representative for all conditions? What about specialists?	We have now provided the sampling matrix and target sample size in Table 1. It would not be feasible to represent each condition whilst keeping the consensus meeting to a manageable size conducive for group discussion. Given our revised plan to now focus on outcomes relevant to all pregnant women with multimorbidity instead of specific conditions, we have opted for representation from a reasonable range of both physical and mental health conditions instead.

24	Consensus meeting: What (qualitative) approach will be used to arrive at consensus? The description provided is vague.	We have added details to describe our planned use of the nominal group technique to arrive at consensus in the Methods: Consensus meeting section. “Nominal group technique will be used to discuss these outcomes. Participants will be asked to contemplate independently whether these outcomes should be included. Each participant will be invited to voice their reasoning in turn using a round-robin format to avoid domination of the discussion by selected few. This will be followed by an open discussion, after which a final anonymous binary vote of yes /no will be conducted for each of these outcomes. Outcomes that received ≥70% yes votes will be included in the final COS.”
25	Discussion: The clinical and research application would need to be defined more clearly.	This has been added to the Discussion section. “The proposed COS will be applicable for observational and interventional studies for pregnant women with pre-existing multimorbidity. Further interventional studies are urgently needed to tackle multimorbidity in pregnancy and reduce the associated adverse outcomes.”
26	Discussion: Inclusion of stakeholders from one country and the serious implications on its generalizability thereof, would need to be addressed as a limitation	We have now removed the limitation of stakeholders to UK only. We have also added the implications of conducting the study in English to the generalisability of the resultant COS to the Discussion Limitation section. “Similarly, the Delphi survey and consensus meeting will be conducted in English. Although efforts will be made to encourage international participation, this may limit the generalisability of the findings to high income countries.”
27	Discussion: The lack of a qualitative step to specifically elicit outcomes especially given the limited search for patient reported outcomes and the lack of qualitative studies aimed at eliciting outcomes in this population, would need be stated as a serious limitation	We have now added a qualitative element to the proposed study using focus groups and outlined this in the Methods section: Stage 2: Focus Groups.

28	Reference list: The references are not reflective of the work done in the area of core outcome sets for pregnancy-associated conditions.	We have now updated the reference list to incorporate more literature in the area of core outcome sets for pregnancy-associated conditions.
	Reviewer 3	
1	Thank you for the opportunity to review for publications. The authors/investigators plan to develop a Core Outcome Set for maternal and offspring outcomes in pregnant women with pre-existing multimorbidity. The authors/investigators state the plan in regards to development of the COS using a 3 step process.	We thank Reviewer 3 for the comments.
2	Abstract: The dates of the study are not stated in the protocol (a requirement for BMJ Open.	The following has been added: Abstract methods section: “The study will be conducted from June 2021 – August 2022.” Manuscript methods section: “The planned start and end dates for the study are June 2021 and August 2022 respectively.”
3	Background: Line 2 the authors should provide the burden of illness, i.e. what is the incidence or rate of multimorbidity issues in the pregnant woman?	This is now added to the 2nd sentence in the first paragraph of the Background section: “Despite an increase in multimorbidity within the general population, there is sparse literature for pregnant women with multimorbidity. Studies in the USA have reported that between 0.8% to 13.9% of hospital births were from women with multiple chronic conditions. Using a list of 79 chronic conditions, our preliminary study found that one in four pregnant women in the UK had active multimorbidity at conception.”

4	Background: Line 6: It is stated that 80% of maternal deaths occur in multimorbidity- what are the 3 or 5 most common causes of death and is there any intervention that can be offered to reduce mortality?	The reference has now been updated with the latest UK national maternal morbidity review, we have added the leading causes of death from this report, however, leading cause of death specific to women with multimorbidity is not available. We have also highlighted the lack of literature for interventions in pregnant women with multimorbidity. “The leading direct cause of maternal death included thrombosis, thromboembolism and maternal suicide; leading indirect cause of death included cardiac diseases, epilepsy and stroke.” “Current observational evidence and interventions focus on single morbidities. There is an urgent need for.... development and evaluation of interventions to improve maternity care for pregnant women with multimorbidity....”
5	Background: c) The 4th paragraph in the background section is all about the "Significance of the study" and not sure if this should be part of this section. More relevant for the discussion section.	This has been moved to the Discussion section.
6	Methods: Scope of the COS- It is stated that long-term outcomes will be included? What does long-term mean (in regards to time frame)? Will such data be available as the outcomes change over time with advances in science?	Long-term outcomes were intended to cover the development of future morbidities, such as gestational diabetes progressing to diabetes or future episodes of depression. However, for clarity, we have removed 'long-term outcomes' as post-partum outcomes would also cover these.

7	Methods: For the offspring outcomes- the authors have made distinction between newborn (first 7 days of life) and neonate (first 28 days of life). However, in medical contexts, newborn or neonate refers to an infant in the first 28 days after birth. The authors need to justify this distinction. Also they are evaluating long-term outcomes of offspring. Can the authors justify or provide the rationale for assessing outcomes across the life span for both mother and her offspring?	We have dropped the newborn period and kept to neonatal period for the first one month of life as suggested. We have added the rationale for why outcomes across the lifespan of mother and offspring are included in the Introduction and Methods section. Introduction: “Besides acute complications (e.g. eclampsia) and chronic complications (progression from gestational diabetes to type II diabetes) for the mother....” Methods: Scope of the COS: “We have included outcomes across the lifespan of offspring to inform observational studies that take a life-course approach. Evidence is emerging that pre-existing maternal morbidities can impact on offspring long-term health in early adulthood.”
8	Methods: What would be the sample size for each of the stakeholder groups?	We have added the target sample size for the focus group, Delphi surveys and consensus meeting in Table 1.
9	Methods: What is the rationale for including journal editors in the stakeholder group?	Journal editors are important stakeholders as they are future implementers of the COS (following the guidance of the COMET handbook), they have been included in other published COS / COS protocol (e.g. https://doi.org/10.1177%2F1753495X18772996).

10	Methods: The number of stakeholders involved for each group not described? How will participant bias/response bias be addressed? How will discrepancies be addressed? Recognizing that there will be potential differences between patient vs. physician preferences or importance in terms of outcomes, how will this issue be addressed?	The target sample size for each stakeholder group is presented in Table 1. The anonymised nature of the Delphi survey and the opportunity to consider the collective response before rerating each outcomes, and the nominal group techniques are methods to address participant and response bias. To address the imbalance between patient and physician preferences, we have now included outcomes preferred by patients but not other stakeholders in the consensus meeting discussions. Methods: Delphi survey: “The response is summarised and fed back to stakeholders anonymously in subsequent rounds. Stakeholders consider the collective views before re-rating the outcomes. This provides a mechanism to reconcile different opinions to reach a consensus.” Methods: Delphi survey: “For further discussion is when: (1) $\geq 70\%$ of all participants rated 4-6 (important but not critical) for an outcome, or (2) when $\geq 70\%$ of patient representatives have rated 7-9 for an outcome but consensus in is not reached.” Methods: Consensus meeting: “Summary scores stratified by stakeholder groups will be presented for outcomes that met the ‘for further discussion’ criteria. Nominal group technique will be used to discuss these outcomes.”
11	Methods: The authors talk about diversity of sample-will patients be selected from different ethnic groups? Will education and SES impact selection of outcomes?	Sampling matrix is provided in Table 1 and includes ethnic minority and socially disadvantaged groups. We will also collect these participants sociodemographic factors. Methods: “To check for representation, the survey will ask for participant characteristics including types of long-term conditions constituting multimorbidity, age, ethnicity, education level and socioeconomic status (patient representatives, as outlined in Table 1), specialty and job roles (health care professionals and researchers).”

12	Methods: The timeline for literature search has not been described? It appears that there is an iterative process and search will be considered until no unique outcomes is identified. Do the authors have an idea of how many years of literature search would have to be conducted based on their previous experience?	We have now changed our literature search strategy to look at multimorbidity in pregnancy in general instead of individual morbidities. As there are limited literature in this topic, we have not limited the years for the search.
13	Methods: The authors list some of the databases that will be searched. Why is Embase excluded? Why limit to English language literature?	We thank Reviewer 3 for the comments and have added EMBASE. We have also removed English language literature as a limitation for our search strategy. Search strategy: "The following databases will be searched: Cochrane library, Medline, EMBASE and CINAHL." Study selection and data extraction: "...No time or language limits will be applied."
14	Methods: Initial list of outcomes provide description of pregnancy outcomes. How will offspring outcomes be reviewed?	Offspring outcomes if identified from the literature search and focus groups will be included in the initial outcome list. Our literature search strategy is broad, using the concept of pregnancy, multimorbidity and offspring. There will also be opportunities for stakeholders to suggest additional outcomes that were not identified from the literature search. Search strategy: "Relevant key search terms will include pregnancy (population and maternal outcomes), multimorbidity (exposure) and offspring (offspring outcomes) derived from previous literature."

VERSION 2 – REVIEW

REVIEWER	Hewawasam, Erandi Women's and Children's Hospital Adelaide, Department of Endocrinology and Diabetes
REVIEW RETURNED	25-May-2021

GENERAL COMMENTS	Authors have made the recommended changes which has significantly improved the manuscript.
--